# Evaluation of Eosinopenia as a SIRS Biomarker in Critically Ill Horses

**DOI:** 10.3390/ani12243547

**Published:** 2022-12-15

**Authors:** María Martín-Cuervo, Luis Alfonso Gracia-Calvo, Beatriz Macías-García, Luis Javier Ezquerra, Rafael Barrera

**Affiliations:** 1Grupo MECIAN, Departamento de Medicina Animal, Facultad de Veterinaria, Campus de Cáceres, Universidad de Extremadura, 10003 Cáceres, Spain; 2Veterinary Teaching Hospital, University of Helsinki, Koetilankuja 1, 00790 Helsinki, Finland; 3Grupo MINVET, Departamento de Medicina Animal, Facultad de Veterinaria, Campus de Cáceres, Universidad de Extremadura, 10003 Cáceres, Spain

**Keywords:** eosinophils, horse, SIRS, prognosis, WBC, lactate

## Abstract

**Simple Summary:**

Eosinopenia has been used as a biomarker of systemic inflammatory response syndrome in critically ill humans. Horses are extremely prone to developing systemic inflammation in different conditions such as endotoxemia. It is for that reason that new biomarkers are needed in horses to rapidly identify the patients that require hospitalization in the intensive care units to minimize unnecessary expenses. The aim of this study was to evaluate eosinopenia as a potential marker of systemic inflammation and prognosis in horses. The results showed lower eosinophil counts in horses affected with systemic inflammation compared to the control group (*p* < 0.05). Horses with eosinopenia were less likely to survive, and hence, eosinophil count could be used as a marker of prognosis and disease.

**Abstract:**

Systemic inflammatory response syndrome (SIRS) is a very common finding in critically ill patients. To accurately identify patients with SIRS and those who need intensive care, several markers have been evaluated, including cortisol, WBC or lactate. It is widely known that a stress leukogram includes eosinopenia as one of its main markers (neutrophilia, eosinopenia, lymphopenia and mild monocytes). It is known that cortisol concentration in plasma is the main stress biomarker and is strongly correlated with the severity of disease in horses. However, it is not possible to measure this parameter routinely in clinical conditions. Hence, in this study it was hypothesized that the eosinophil count could be a reliable parameter to identify critically ill horses. Horses included in this study were divided into three groups: Group A (sick horses received at the Emergency Unit which did not fulfil the criteria for SIRS), Group B (horses that meet two or more criteria for inclusion in the definition of SIRS) and a control group of healthy horses. In this study, horses with SIRS showed lower eosinophil counts than healthy horses. Moreover, non-surviving horses exhibited lower eosinophil counts than survivors. Eosinopenia could be used to identify horses with SIRS and can be useful as a prognostic marker.

## 1. Introduction

Systemic inflammatory response syndrome (SIRS) in horses has been associated with mortality and with a need for intensive care in humans [1] and horses [2]. The definition of SIRS in adult horses includes two or more of the following abnormalities: fever or hyperthermia (>38.6 °C), tachycardia (>60 beats/min), tachypnea (respiratory rate >30 breaths/min) and white blood cell count (WBC) >12,500 cells/μL or <4500 cells/μL and 10% band neutrophils [3]. Although there is not a consensus on the exact use of the terms sepsis, endotoxemia and SIRS in horses [4] and sometimes the terms are used interchangeably, the more accepted concept is that endotoxemia is a SIRS-associated complication related to the presence of endotoxins, which are the lipopolysaccharide components of the cell wall of Gram-negative microorganisms. Regardless of the terminology used, it is important to provide tools for the prompt recognition and diagnosis of SIRS to equine clinicians because early diagnosis and treatment may lead to a reduction in both mortality and morbidity [5]. Systemic inflammation causes deleterious effects in the host due to massive release of inflammatory mediators including cytokines, eicosanoids, complement activation factors and stress hormones. Cortisol in horses is a valuable marker of prognosis and severity of SIRS [6] and it is related to the stress leukogram [7]. Eosinopenia is part of the canonical stress leukogram that responds to cortisol release. Cortisol and other stress indicators, such as “heat shock protein 72” (HSP72) and beta-endorphins, have been used as prognosis markers. The risk of death also appears to be higher in horses affected by gastrointestinal lesions with high circulating concentrations of epinephrine and cortisol, indicating a high degree of activation of the sympathetic system in horses with colic [8].

However, cortisol cannot be measured in clinical conditions and it would be helpful to outline other parameters related to the degree of stress. Patients with SIRS can develop leukopenia in the first stages of the disease, which is difficult to identify based on the stress leukogram [9]. Therefore, it is difficult to establish a correlation between the severity of the disease and the WBC count.

Other biomarkers have been used in critically ill horses with variable results. Lactate is the more widely used biomarker in critically ill horses and remains the more prominent parameter used as a prognosis marker [2,10]. Other biomarkers such as procalcitonin [11] or C-reactive protein [12] have been studied in equine patients but its clinical utility is not as promising as in human medicine. Serum amyloid A (SAA) seems to be useful for monitoring treatments [13] but it does not allow one to discriminate between horses with SIRS and horses with local inflammation [14].

On the other hand, eosinopenia typically accompanies the response to acute infection in human medicine [10]. This marked reduction in the number of circulating eosinophils in acute infection was first described by Zappert in 1893 [11] and was used during the first quarter of the last century as a diagnostic sign [12]. Taking the fact that eosinopenia is part of the normal response to stress into account [13], it could be assumed that eosinopenia in horses is a secondary response to the heavy stress caused by systemic inflammation [8,14]. The value of this ancient marker of acute infection in humans was tested by Gil et al. [15]. It is well known that corticosteroids induce eosinopenia [16] and recently, it has been used in humans as biomarker of diagnosis [17] and prognosis [18].

Because eosinophil count has been associated with the amount of circulating cortisol, we hypothesized that horses with SIRS may present lower eosinophil counts than sick horses without SIRS. To our knowledge, however, there is no earlier study testing the value of eosinopenia in the diagnosis of SIRS in critically ill horses. The aim of the present study was to assess the value of eosinopenia in distinguishing critically ill horses from other equine patients on admission.

## 2. Materials and Methods

A prospective study was performed in which horses admitted to the emergency services at the Veterinary Teaching Hospital of the University of Extremadura during 2021 were included. Horses younger than 2 years were excluded from the study. Informed consent was not demanded because this observational study did not require any deviation from routine medical practice. Moreover, a control group of 31 healthy horses was included.

### 2.1. Animals

At the time of admission, the age, gender, principal diagnosis, and vital signs (body temperature, heart rate, respiratory rate) were recorded for each patient (Appendix A).

Patients were classified as having SIRS if they met two or more clinical signs of the SIRS criteria.

To assess the value of eosinopenia as a marker of SIRS, the eosinophil cell count, WBC count and lactate were compared between the three different groups, which included:-Group A: Sick horses received in the Emergencies Unit that did not fulfil the criteria for SIRS.-Group B: patients received in the Emergencies Unit that met two or more criteria for inclusion in the definition of SIRS.-Control group: healthy horses without clinical signs of disease and normal blood tests. These horses were admitted in the hospital for elective surgeries and the blood sample was taken as part of the regular protocol.

Survival was estimated as discharge from the hospital after the treatment. Horses euthanized due to economic restraints were not included.

### 2.2. Blood Parameters

The following laboratory parameters were systematically recorded on admission as part of the regular protocol: white blood cell count (Appendix A), the eosinophil cell count, lactate, fibrinogen, creatinine, urea, hematocrit, total proteins, albumin, and total bilirubin. Blood samples were obtained by venipuncture of the jugular vein on admission.

Blood samples were collected in microtubes containing ethylenediaminetetraacetic acid anticoagulant (EDTA). The white blood cell count was performed by a semiautomatic electronic blood cell counter (Sysmex F-800). Moreover, the eosinophil cell count was performed by a manual Diff by classifying 200 WBCs on a blood smear to determine the percentage of each type of WBC present. The percentage of eosinophils was multiplied by the total WBC count/μL to obtain the absolute count of this WBC type. The leukocyte differential count was performed manually to detect significant toxic changes in neutrophils. To determine the lactate level, blood samples were drawn into green-top vacutainer tubes containing lithium-heparin as the anticoagulant. Plasma lactate was measured by immunoturbidimetry using a Clinical Chemistry Analyzer (Saturno 100 Vet Crony^®^ Instruments, Rome, Italy). The limit of detection was 0.071 mg/dl. Plasma fibrinogen was collected into blue-top vacutainer tubes containing sodium citrate and was measured by a thrombin coagulation technique (Hemofibrin-kit^®^, Laboratorio Gernon S.A., Barcelona, Spain).

### 2.3. Statistical Analysis

Descriptive statistics were presented as the mean ± standard deviation. To compare the eosinophil cell count, lactate concentrations and WBC count between groups, a Shapiro–Wilk test was used to check the data distribution and, in light of the non-gaussian distribution, a Kruskal–Wallis one-way analysis was used to assess differences between values. A Mann–Whitney rank sum test was used to evaluate the patient’s outcome. A *p* value < 0.05 was considered statistically significant. All statistical analyses were performed using Prism version 9 (San Diego, CA, USA).

## 3. Results

During the study period, 49 patients were admitted to the intensive care unit and 37 fulfilled the inclusion criteria. The horses were divided into two groups: 16 sick horses received in the Emergency Unit which did not fulfil the criteria for SIRS that were included in Group A and 21 horses that meet two or more criteria for inclusion in the definition of SIRS that were included in Group B (Table 1). A control group with 31 healthy horses without clinical signs of disease and normal blood tests was included.

### 3.1. Eosinophils, WBC and Lactate Concentrations Regarding SIRS

There were no significant differences in the WBC counts between groups (Table 2). There were significant differences in the lactate concentration between groups, with higher values in the clinical groups of interest (Control group: 1.3 ± 0.6 mmol/L; Group A: 2.1 ± 0.9 mmol/L; Group B: 5.8 ± 6.3 mmol/L) (Table 3). There were significant differences in the eosinophil counts between the control group and Group B, with lower counts in the SIRS group than in healthy horses (Control group: 238.1 ± 372.8 cell/ μL; Group A: 103.7 ± 155.5 cell/ μL; Group B: 12.4 ± 31.8; *p* > 0.001) (Table 4).

### 3.2. Outcome

The overall survival rate was 54%, with 15 non-survivors in the SIRS group (71.43%), and 4 non-survivors in the non-SIRS group (25%). There were no significant differences in the WBC count (Table 5) and lactate (Table 6) between survivors and non-survivors (WBC: survivors: 8802 ± 2664 cell/ μL; non-survivors: 7258 ± 4450 cell/ μL. Lactate: survivors 3.0 ± 2.2 mmol/L; non-survivors: 5.7 ± 6.9 mmol/L). There were significant differences in the eosinophil counts between the survivors and the non-survivors, as the counts were higher in the survivor group (Survivors: 67.9 ± 120.1 cell/ μL; non-survivors: 3.8 ± 14.7 cell/ μL) (Table 7).

## 4. Discussion

The eosinophil count was higher in the survivors than in non-survivors and higher in the healthy horses than in sick horses. Eosinopenia has been used in human medicine as an indicator of sepsis [19,20,21], as an outcome predictor [18,22,23] and in patients with abdominal pain [24]. The mechanisms that control eosinopenia in acute inflammation, also considered acute stress, include the adrenal release of glucocorticoids and epinephrine [10]. Additionally, the initial eosinopenic response to acute infections can be interpreted as being the result of peripheral sequestration of circulating eosinophils. In addition, chemotactic substances released during acute inflammation such as C5a and fibrin fragments may contribute to eosinophil migration and sequestration, in conjunction with the inflammatory response itself [10,25,26].

In view of all the previously mentioned work, it is easily explained that animals under stress and, consequently, with higher concentrations of catecholamines, present the lowest eosinophil counts. As demonstrated in previous studies, cortisol is a good prognostic marker, as its concentrations are higher in non-surviving horses or in those affected by more severe diseases [8]. The measurement of cortisol in the equine clinic is not performed routinely given its complexity and its high economic cost, and has only been used in experimental studies. Therefore, the eosinophil count (which is performed routinely in the equine clinic, both manually and automatically), may be a good alternative, since there seems to be a relationship between their count and the level of circulating cortisol. The results showed that eosinophil counts can help distinguish healthy horses from sick horses. Unfortunately, it does not seem to be a very useful parameter to accurately differentiate horses with SIRS from those without systemic inflammation. Similar results have been observed in human patients with sepsis [27].

In this study, lactate levels have shown to be a more reliable marker than WBC and eosinophil counts for identifying horses with SIRS. The increase in lactate concentration generally occurs when the tissular demand for energy exceeds the availability of oxygen in blood. Plasma lactate is currently being used to assess the degree of ischemia and tissue perfusion in critically ill human patients [28]. Its usefulness as a prognostic marker has been demonstrated, both in horses with digestive disorders [29,30,31] and in neonatal foals suffering from septic processes [32,33]. It is also useful to establish the severity of the process as well as to evaluate the response to treatment in critically ill horses [34,35]. Plasma and peritoneal lactate concentration are the most used parameters in determining if a horse requires medical or surgical treatment when suffering from gastrointestinal diseases [36,37].

WBC count provides information about immunity status and it has been used for the evaluation of inflammation. Leukocytosis consists of an increase in the number of leukocytes in the peripheral circulation above the reference ranges. Its causes include bacterial and viral infections, traumatic injuries, burns, stress, corticosteroid administration [38], immune-mediated processes, epinephrine release, excess production (bone marrow neoplasms), abnormal migration or inability to migrate (adhesion deficiency) and alterations in their functional capacity [39]. The term leukopenia refers to a decrease in the number of circulating leukocytes below the limits that are considered physiological. Its causes include massive infections (bacterial or viral), endotoxemia, severe diseases and failure of synthesis (alterations of the bone marrow) [40,41]. In this study, the WBC count did not differentiate healthy from sick horses or distinguish those with systemic inflammation from those with less severe disease. These results could be explained by the diversity of the diseases in the patients included in Groups A and B, which consisted of disorders characterized by leukocytosis, such as peritonitis, as well as others in which leukopenia is more characteristic, such as acute gastrointestinal processes.

Lactate did not provide relevant information about the outcome. There are many causes of increased L-lactate concentrations in blood and other biological fluids. Two types of lactic acidosis have been described based on different causes. Lactic acidosis type A is the result of tissue hypoxia, secondary to hypoperfusion, decreased oxygen concentration in arterial blood, or tissue problems in oxygen mobilization. Type B hyperlactacidemia is the consequence of mitochondrial dysfunction, alterations in carbohydrate metabolism, or a decreased rate of lactate clearance [28]. The most common and important causes of hyperlactacidemia are alterations of tissue perfusion and hypoxia. However, the increase in plasma lactate concentration can occur in critically ill patients suffering from diseases where oxygen transport to the tissues is normal. When lactate production from hypoxic tissues exceeds the rate of elimination through the kidneys and liver, its concentration increases in the blood [42]. The broad diagnosis included in this study could have biased the results of lactate concentration, as horses with severe asthma or dehydration can show increases in lactate with no other signs related to poor prognosis.

WBC count has been demonstrated to be not useful as a prognostic marker. In human medicine, new research found that it was more relevant to analyze WBC trajectories than the count on admission as a prognostic marker [43]. In horses, it seems to be more interesting to include the detection of band cells or the neutrophil toxic changes than the total WBC count [7].

Low eosinophil counts have shown to be more useful as a prognostic marker than the WBC count in several diseases [44,45]. In human medicine, eosinopenia is considered a reliable prognostic marker in acute ischemic stroke [46], coronavirus [47], urticaria [48], non-cardiac vascular surgery [49] and acute myocardial infarction [50]. There is also an association between persistent eosinopenia and high mortality in aged hospitalized patients [51]. However, despite the extensive bibliography that exists in human medicine, to the authors’ knowledge, there are no studies on eosinopenia as a prognostic marker in horses. In this study, low eosinophil counts have been observed in horses that do not survive, coinciding with previous findings reported in human medicine. As the eosinophil counts are part of the protocol in intensive care units and are routinely included in modern hematological equipment, in light of our findings, more attention should be paid to this parameter. Furthermore, thanks to their characteristic morphology, horse eosinophils are very easy to distinguish, making this method an effective alternative for this type of count.

The limitations of this study are the small number of animals included and the heterogeneity of the diseases analyzed. More studies are needed to evaluate the progression of eosinopenia in hospitalized critically ill patients and to establish a cut-off value for this marker. Caution must be taken when interpreting these results, since under normal conditions, the eosinophil count is quite low, so the values obtained should be interpreted together with other more robust parameters such as lactate.

## 5. Conclusions

The results obtained in this work show that the eosinophil count can be useful for differentiating healthy from sick horses, although it is not a sensitive marker of SIRS. Eosinopenia can be used as a prognostic marker in critically ill horses. However, these results should be interpreted with caution, and it is advisable to include other parameters in decision making.

## Figures and Tables

**Table 1 animals-12-03547-t001:** Demographics and laboratory data of horses on admission. The diagnosis and outcome of each patient is reported (Groups A and B).

Horse	Group	Lactate (mmol/L)	Eosinophils (Cell/μL)	WBC (Cell/μL)	Outcome	Diagnoses
1	A	1	162	5400	Survival	Diarrhea
2	A	2.5	428	10,700	Survival	Medical colic
3	A	1.2	0	8900	Survival	Medical colic
4	A	1.8	0	6300	Survival	EGUS
5	A	1.7	320	6400	Survival	Medical colic
6	A	2.2	0	12,200	Survival	Colon displacement
7	A	3.7	0	12,100	Non-survival	Inguinal herniation
8	A	2.8	428	10,700	Non-survival	Inguinal herniation
9	A	0.8	102	10,200	Survival	Severe asthma
10	A	3	0	10,900	Survival	Esophageal obstruction
11	A	3.7	0	5200	Survival	Colon displacement
12	A	2.3	57	5700	Non-survival	Laminitis
13	A	1.7	0	10,800	Survival	Laminitis
14	A	-	162	8100	Survival	Severe asthma
15	A	2.2	0	9100	Survival	Severe asthma
16	A	1.3	0	13,000	Non-survival	Lymphoma
17	B	7.2	0	11,300	Survival	Enteritis
18	B	5.5	0	1100	Non-survival	SI volvulus
19	B	4.2	0	7600	Survival	Medical colic
20	B	2.4	0	10,600	Survival	Colon displacement
21	B	3.1	0	8800	Non-survival	Peritonitis
22	B	2.5	0	12,800	Non-survival	Peritonitis
23	B	2.1	0	9300	Non-survival	Inguinal herniation
24	B	4.5	72	5400	Survival	Medical colic
25	B	12.2	0	3000	Non-survival	SI volvulus
26	B	1	0	4500	Survival	Fecaloma
27	B	4.1	0	9100	Non-survival	SI obstruction
28	B	3.5	76	3888	Non-survival	Rectal rupture
29	B	2	112	7840	Survival	Colon displacement
30	B	30	0	15,400	Non-survival	Peritonitis
31	B	1.4	0	2500	Non-survival	Colitis
32	B	6.3	0	5700	Non-survival	SI volvulus
33	B	5.7	0	2100	Non-survival	Dystocia
34	B	9.1	0	3800	Non-survival	Inguinal herniation
35	B	1	0	4500	Non-survival	Medical colic
36	B	5.8	0	14,000	Survival	Enteritis
37	B	8.6	0	10,600	Survival	Inguinal herniation

Note: - is for missing data; EGUS: equine gastric ulcer syndrome; SI: small intestine.

**Table 2 animals-12-03547-t002:** Results of WBC counts (cell/ μL) in all groups.

Group	N	Median	25%	75%
Control Group	31	7900	7400	8700
Group A	16	9650	6325	10,875
Group B	21	7600	3800	10,600

Note: there is not a statistically significant difference (*p* = 0.233).

**Table 3 animals-12-03547-t003:** Results of lactate levels (mmol/L) in all groups.

Group	N	Median	25%	75%
Control Group	31	1.2 ^a,b^	1.0	1.4
Group A	16	2.2 ^a^	1.3	2.8
Group B	21	4.2 ^b^	2.2	6.7

Note: ^a,b^: there is a statistically significant difference (*p* ≤ 0.001).

**Table 4 animals-12-03547-t004:** Results of eosinophil counts (cell/μL) in all groups.

Group	N	Median	25%	75%
Control Group	31	160 ^a^	0.0	296.0
Group A	16	0	0.0	162.0
Group B	21	0 ^a^	0.0	0.0

Note: ^a^: there is a statistically significant difference (*p* ≤ 0.001).

**Table 5 animals-12-03547-t005:** Results of WBC counts (cell/ μL) depending on survival. Control group was not included.

Group	N	Median	25%	75%
Survivors	20	9000	6325	10,775
Non-survivors	18	5700	3400	11,400

Note: there is not a statistically significant difference (*p* = 0.223).

**Table 6 animals-12-03547-t006:** Results of lactate levels (mmol/L) depending on survival. Control group was not included.

Group	N	Median	25%	75%
Survivors	20	2.2	1.7	4.2
Non-survivors	18	3.5	2.2	6.0

Note: there is not a statistically significant difference (*p* = 0.120).

**Table 7 animals-12-03547-t007:** Results of eosinophil counts (cell/ μL) depending on survival. Control group was not included.

Group	N	Median	25%	75%
Survivors	20	0	0.0	109.5
Non-survivors	18	0	0.0	0.0

Note: there is a statistically significant difference (*p* = 0.039).

## Data Availability

Not applicable.

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
