# Peer review of "Evaluation of Eosinopenia as a SIRS Biomarker in Critically Ill Horses"

_animals, 2022, doi:10.3390/ani12243547_

Round 1
Reviewer 1 Report
General comments
This report addresses if eosinopenia is a valuable marker for horses with SIRS for the first time in horses, a marker used in human medicine. The topic if highly interested in equine medicine as endotoxemia is a common complication in this species and the prognosis is poor.
I commend the authors for the manuscript presented. The length and structure are correct, and in general terms, it is effortless to understand. Nevertheless, there are some concerns that authors need to address before publication.
Mayor concerns
None.
Minor concerns
1. Introduction
- Page 1-2: “Currently, there is not a consensus…..Hence, it is important morbidity (5).”
I don´t understand the connexion of no having a consensus in the term and the importance of making an early diagnose. Firstly, this report does not address the effort performed or not by the scientific veterinary medical institution, and thus, I would omit that sentence in any case. Secondly, the need of an early diagnose may be supported by the fact that endotoxemia increase the risk of morbidity and mortality. Please, change this part of the introduction.
- Page 2: Authors only report cortisol as a biomarker in horses with SIRS, but there are more. Please include the other biomarkers reported. Most of them are also more difficult to address and the useful of using eosinophils still being an advantage
- Page 2: Authors highlight in the result section the lactate levels, but the reason of this is not clear for me. If these two parameters are the special interest, the introduction should mention both.
- Page 2:” On the other hand, …” in this paragraph there are two “well known”. Please change one of them.
2. Material and methods
- Authors clarify the use of the clinical data (owners’ agreement?).
- Animals:
o Please include categorization of the gender and the units of the other parameters.
o Eosinophil is a WBC. I would change to: Total WBC; and I would include the rest of these cells measured (neutrophils, etc).
o The reason for comparison of eosinophils with lactate and WBC is not clear.
o Please move: “Patients were classified...” to the next paragraph.
o Group A: please delete “were included”.
o Group B: these horses were received at the Emergency Unit? I yes, I would use the same sentence than in group A, just highlighting the difference.
o Control group: please delete “a control group of”.
o In general, please consider changing the name of group A and B as with this format the reader must remember which met criteria or not when analysing tables. A suggestion may be: Group non-SIRS; Group SIRS; Group CTL (or Healthy) as authors used in 3.2.
- Blood parameters: This is described in “Animals”. Please consider to moved the blood parameters to this section or to change this two subtitles. A suggestion may be: “Animals and data recorded” with “Blood parameters quantification”.
- Statistic:
o Authors use KW One Way, thus I interpret they did not have normality. If this correct, mean and standard deviation are not proper. Using median and s.e.m may be more correct. Also, percentiles may be appropriate. Actually, authors use it in the Table 2,3, and 4 (without describing it in this section).
o Please cite the Normality test before comparisons with KW One Way and clarify if there is or not normality.
o Mann W. is used to assess if there are differences in total WBC, Lactate and Eos. between horses sick that survival or not. Please, rewrite it properly as survival tests usually are others.
o Did the authors calculate the sample size required?
3. Results
- The reason of showing WBC and lactate is not clear in the manuscript.
- First line: change but only (which may have a negative interpretation) by “and”.
- Table 1: This is not results of “mean and sd”. This is a table with all the data. Please correct the legend. In addition, gender and age may be included. This table should be “demographic and clinical data of horses at the time of admission, including in the study…”
- 3.1:
o “SIRS. Please extend this subtitle to highlight the results provided in it.
o Please unified the 3 tables in 1. Unified the number of decimals in all the manuscript and tables.
o Decimals in English are marked with a point. Please review all the manuscript and correct it.
o Please do not use decimals in total WBC; and use 1 decimal for lactate and eosinophils.
o “There were significant differences in 129 the eosinophil count between the control group and group B being higher the counts in 130 healthy horses”. It may be better to highlight that in group B is lower…., which is the hypothesis.
- 3.2: Pleas, unified these tables in one.
4. Discussion
- The first paragraph should be moved to the introduction.
- Please, star this section with the main finding in a paragraph followed by the discussion of eosinopenia.
- Unified the discussion of lactate and WBC.
Reviewer 2 Report
The results can be presented more clearly, since having the values inserted in the text is makes it less readable. this part can be improved in the way it is written.
Author Response
The results can be presented more clearly, since having the values inserted in the text is makes it less readable. this part can be improved in the way it is written.
The authors thank the reviewers for the time taken to read and suggest changes in our manuscript. We have tried to make it more easily readable.
Reviewer 3 Report
Thank you for submitting your manuscript. The use and importance of eosinophils in severe diseases and SIRS is quite interesting, and the results of the study are important to provide diagnostic and prognostic markers for SIRS in horses.
I would suggest having the paper revised by a native English speaker, because there are some minor spell and grammar mistakes that should be corrected.
Could you please move the paragraph about WBC and eosinophils (lines 224-241) together with the first discussion on WBC (lines 195-209)? I think this way the discussion would be more fluid to read.
Lines 193-194: please specify if you are referring to gastrointestinal diseases
Line 223: please check what "CITA" means
Lines 254-255: please check the sentence, I think you forgot to delete part of the lines provided in the template.
Author Response
x
Thank you for submitting your manuscript. The use and importance of eosinophils in severe diseases and SIRS is quite interesting, and the results of the study are important to provide diagnostic and prognostic markers for SIRS in horses.
The authors thank the reviewers for the time taken to read and suggest changes in our manuscript.
I would suggest having the paper revised by a native English speaker, because there are some minor spell and grammar mistakes that should be corrected. Thank you for your comments. The paper has been reviewed and corrected.
Could you please move the paragraph about WBC and eosinophils (lines 224-241) together with the first discussion on WBC (lines 195-209)? I think this way the discussion would be more fluid to read. We have changed the discussion as suggested.
Lines 193-194: please specify if you are referring to gastrointestinal diseases. The clarification has been included.
Line 223: please check what "CITA" means. It is has been removed
Lines 254-255: please check the sentence, I think you forgot to delete part of the lines provided in the template. The sentence has been removed

Reviewer 4 Report
Exploratory studies on the diagnostic feasibility of eosinopenia in the treatment of horses are important for veterinary practice, but must be carried out at a high methodological level and have a weighty evidence base. Unfortunately, the studies performed have methodological errors that do not allow us to adequately evaluate the data obtained and draw certain conclusions. It is obvious that the amount of eosinophils in the blood of horses normally varies in a certain interval and has the character of a normal distribution. This thesis is easy to verify with statistical processing of leukogram data of healthy horses used in the study. Even with a relatively low content of the number of eosinophils in the bloodstream of horses (0.2 x 109/ l), it is hardly possible to talk about the complete absence of these cells in most patients of groups A and B (Table 1).
It is known that the cause of eosinopenia can be several factors, including taking glucocorticoids; stress accompanied by cortisol release; severe allergic reactions; infections, sepsis and injuries. Therefore, without monitoring the level of cortisol in the blood of horses, it is not correct to study its relationship with the number of eosinophils. It is necessary to provide more complete data on the clinical examination of patients, as well as to discuss the correspondence of the anamnesis to the diagnosis and the results of treatment.
Author Response
Exploratory studies on the diagnostic feasibility of eosinopenia in the treatment of horses are important for veterinary practice, but must be carried out at a high methodological level and have a weighty evidence base. Unfortunately, the studies performed have methodological errors that do not allow us to adequately evaluate the data obtained and draw certain conclusions. It is obvious that the amount of eosinophils in the blood of horses normally varies in a certain interval and has the character of a normal distribution. This thesis is easy to verify with statistical processing of leukogram data of healthy horses used in the study. Even with a relatively low content of the number of eosinophils in the bloodstream of horses (0.2 x 109/ l), it is hardly possible to talk about the complete absence of these cells in most patients of groups A and B (Table 1).
It is known that the cause of eosinopenia can be several factors, including taking glucocorticoids; stress accompanied by cortisol release; severe allergic reactions; infections, sepsis and injuries. Therefore, without monitoring the level of cortisol in the blood of horses, it is not correct to study its relationship with the number of eosinophils. It is necessary to provide more complete data on the clinical examination of patients, as well as to discuss the correspondence of the anamnesis to the diagnosis and the results of treatment.
The authors thank the reviewers for the time taken to read and suggest changes in our manuscript.
Two supplementary files have been added including clinical and additional laboratory data. These data have also been mentioned in the results and discussion sections.

Round 2
Reviewer 4 Report
The authors significantly improved the quality of the article, but did not solve the main problem of the methodological plan - the actual determination of the number of eosinophil cells in 1 μL of blood. .The analyzer takes into account only the approximate percentage of eosinophil cells relative to the number of leukocytes (WBC), therefore, with a small number of eosinophils in the sample (1-5 cell/ μL ), we get an approximate estimate of 0, which is duplicated in further calculations in the data of Table 1. Such an inconspicuous system error leaves its imprint on further calculations and leads to an abnormally high error of average values (>150%), which also indicates a methodological error.
Group A 16 60.93 ± 160.74; Group B 21 261.57 ±480.43
I can recommend the authors to use a more accurate method of counting eosinophil ceils in the blood, including even a routine microscopy method that confirms the presence of these cells in almost any blood smear.
Author Response
Actually, the eosinophil count by direct visualization (and differential leukocyte count) was done because it is part of our standard protocol. The reason to remove this information from the article was because the hematological analyser (Sysmex F-800) is validated in horses with good correlation. For that reason we considered not needed included this information.
Pastor J, Cuenca R, Velarde R, Marco I, Viñas L, Lavín S. Evaluation of a haematological analyser (Sysmex F-800) with equine blood. Zentralbl Veterinarmed A. 1998 Mar;45(2):119-26. doi: 10.1111/j.1439-0442.1998.tb00807.x. PMID: 9591475.
But if it considered needed we can't included the following sentence:
The eosinophil cell count was performed by a manual Diff-Quick dye by classifying 200 WBCs on a blood smear to determine the percentage of each type of WBC present. The percentage of eosinophils was multiplied by the total WBC count/μl to obtain the absolute count of these WBC type. Leukocyte Differential Count was performed manually in order to detect significant toxic changes in neutrophils.